# Comparison of System of Rice Intensification Applications and Alternatives in India: Agronomic, Economic, Environmental, Energy, and Other Effects

Rapolu Mahender Kumar [1,*], Padmavathi Chintalapati [1], Santosha Rathod [1], Tapeshwar Vidhan Singh [1], Surekha Kuchi [1], Prasad Babu B. B. Mannava [1], Patharath Chandran Latha [1], Nethi Somasekhar [1], Nirmala Bandumula [1], Srinivas Prasad Madamsetty [1], J. V. N. S. Prasad [2], Shanmugam Vijayakumar [1], Dayyala Srinivas [1], Banugu Sreedevi [1], Mangal Deep Tuti [1], Melekote Nagabhushan Arun [1], Banda Sailaja [1] and Raman Meenakshi Sundaram [1]

1   ICAR-Indian Institute of Rice Research, Rajendranagar, Hyderabad 500 030, India;
    chintalapatipadmavathi68@gmail.com (P.C.); santoshagriculture@gmail.com (S.R.);
    vidhan_singh@yahoo.com (T.V.S.); surekhakuchi@gmail.com (S.K.); mbbprasadbabu@gmail.com (P.B.B.B.M.);
    lathapc@gmail.com (P.C.L.); nssekhar@hotmail.com (N.S.); nirmalaicar@gmail.com (N.B.);
    ms.prasad1@icar.gov.in (S.P.M.); vijitnau@gmail.com (S.V.); seenu290@gmail.com (D.S.);
    sreedevi.palakolanu@gmail.com (B.S.); mangal.iari@gmail.com (M.D.T.); arun_tulasi2011@yahoo.in (M.N.A.);
    bandasailaja@gmail.com (B.S.); rms_28@rediffmail.com (R.M.S.)
2   All India Coordinated Research Project on Dryland Agriculture, ICAR-Central Research Institute for Dryland
    Agriculture, Santhosh Nagar, Hyderabad 500 059, India; jasti2008@gmail.com
*   Correspondence: kumarrm21364@gmail.com; Tel.: +91-040-24591236

**Abstract:** Initial evaluations of the System of Rice Intensification in India and elsewhere focused mainly on its impacts on yield and income, and usually covered just one or two seasons. Researchers at the ICAR-Indian Institute of Rice Research have conducted a more comprehensive evaluation of SRI methods over six years (six wet and six dry seasons), comparing them with three alternatives: modified, partially mechanized SRI (MSRI) to reduce labor requirements; direct-seeded rice (DSR) as an alternative method for growing rice; and conventional transplanting of rice with flooding of fields (CTF). Grain yield with SRI methods was found to be about 50% higher than with CTF (6.35 t ha$^{-1}$ vs. 4.27 t ha$^{-1}$), while the MSRI yield was essentially the same (6.34 t ha$^{-1}$), 16% more than with DSR (5.45 t ha$^{-1}$). Water productivity with SRI methods was 5.32–6.85 kg ha-mm$^{-1}$, followed by 4.14–5.72 kg ha-mm$^{-1}$ for MSRI, 5.06–5.11 kg ha-mm$^{-1}$ for DSR, and 3.52–4.56 kg ha-mm$^{-1}$ for CTF. In comparison with CTF, SRI methods significantly enhanced soil microbial populations over time: bacteria by 12%, fungi by 8%, and actinomycetes by 20%. Biological activity in the rhizosphere was also higher as indicated by 8.5% greater dehydrogenase and 16% more FDA enzymes in soil under SRI management. Similarly, an indicator of soil organic matter, glucosidase activity, was 78% higher compared to CTF. SRI enhanced the relative abundance of beneficial microbial-feeding nematodes by 7.5% compared to CTF, while that of plant-pathogenic nematodes was 7.5% lower under SRI. Relative to conventional methods, SRI management reduced GHG emissions by 21%, while DSR reduced them by 23%, and MSRI by 13%, compared to standard rice-growing practice. Economic analysis showed both gross and net economic returns to be higher with SRI than with the other management systems evaluated. While the six-year study documented many advantages of SRI crop management, it also showed that MSRI is a promising adaptation that provides similar benefits but with lower labor requirements.

**Keywords:** climate resilience; crop establishment methods; greenhouse gas emissions; mechanical transplanting; rice; soil nematodes; system of rice intensification

## 1. Introduction

Rice is a principal food crop for about half of the people in the world. Globally, rice production occupies an area of nearly 165 million hectares, producing 787 million tonnes of paddy rice in 2021 [1], representing an average productivity of 4.67 t ha$^{-1}$. In India, the total area under rice cultivation is 46.4 m ha, with a total production of 186 million tonnes of paddy rice (130.3 million tonnes of milled rice) and an average yield of 2.70 t ha$^{-1}$ [2].

Nearly 90% of the world's rice is produced in Asia, with India as a significant contributor. It is India's number one staple food crop and contributes significantly to the livelihood of most people. In recent years, the area under rice cropping has been somewhat decreasing, however, due to urbanization, migration of labor from agriculture to non-agricultural sectors, increasing input and labor costs, and growing water shortages, all seriously threatening the continuing cultivation of rice [3].

Among cereals, irrigated lowland rice with continuous flooding consumes the most water of any crop, with considerable wastage of water. The challenge is to develop technologies that maintain or increase rice production with reduced water consumption. Unfortunately, the world's rice production is in crisis, and India is not an exception, with declining cultivated areas, variable production, stagnant yield, and escalating input costs. Hence, growing more rice with fewer input requirements is needed, particularly with less water requirements [4].

In recent years, the System of Rice Intensification (SRI) method has been gaining wider acceptance worldwide due to its greater yield and lower costs as well as more efficient utilization of water. It has demonstrated positive results in China and India and more than 60 other countries in Asia, Africa, and Latin America [5].

As transplanting rice seedlings requires nearly 25% of the labor for irrigated rice production [6], finding ways to reduce this requirement through mechanization is very desirable. The labor requirements for SRI methods vary considerably, affected by skill, experience, and other factors. SRI transplanting involves only 10–20% as many seedlings, but these need to be planted with greater care and precision, so SRI's greater labor requirements, at least initially, impede its adoption. Farmers will benefit from labor-saving cultivation methods that also reduce their costs of production if this does not diminish their output of grain.

The operations of transplanting, weeding, and harvesting require about 60–80% of the labor needed for rice production [7]. Harvesting, both the cutting and threshing of grain, is already mechanized or can be. Unfortunately, the mechanization of transplanting and weeding for SRI is less advanced. In particular, transplanting by machine rather than manually could greatly reduce the time needed for crop establishment, and this would enhance the profitability of rice production [8]. The partial mechanization of SRI (MSRI) thus deserves thorough evaluation, comparing this with manual transplantation.

Another option for crop establishment is the direct seeding of rice (DSR), which is becoming more popular among farmers in India as it requires less labor and is less costly [9]. Yields with DSR can be comparable with those of transplanted rice if there is good seed germination and establishment. Thus, this option also warrants evaluation. Accordingly, field experiments were conducted in both wet and dry seasons to assess these alternative methods of rice production in terms of grain yield, energy efficiency, greenhouse gas emissions, water productivity, impact on the soil biota, and economic profitability.

Changing from CTF to SRI, MSRI or DSR crop management will change the soil food web within rice ecosystems, favoring those species that prefer a more aerobic, less water-saturated soil environment. This can have varying effects on crop performance. For example, SRI's water management that alternates wet and dry rice paddies could lead to an increase in the populations of pathogenic plant-parasitic nematodes, such as root-knot and lesion nematodes, that thrive under aerobic conditions. On the other hand, it will increase mycorrhizal fungi that are beneficial for plant growth [10]. Applying more organic matter and fewer chemical inputs to the soil can support the expansion of populations of various soil microbes, higher levels of soil enzyme activity, more nutrient availability, and

an enhanced nutrient pool for plant roots [11]. The positive and negative effects of SRI management on the soil biota should therefore be considered.

To assess the impacts of agricultural management on the soil biota, it is important to track practices over several years, to understand their longer-term effects on the soil food web. Some studies have already indicated that SRI practices foster more favorable soil-microbe–plant relationships compared to CTF [12]. We decided to track changes particularly in the populations of soil nematodes, both harmful and beneficial species, under both SRI and CTF management since these microfauna are very important for the health and growth of rice plants.

The aerobic soil conditions maintained under SRI management encourage improved root health and function, leading to greater root growth, and favor the development and activity of a larger, more diverse soil microbiota [13]. Indicators such as the populations of bacteria, fungus, and actinomycetes and the levels of soil enzyme activity were thus monitored in this six-year study to comprehend what alterations in soil microbiology might result over time from SRI vs. conventional methods for rice cultivation.

The studies reported here were undertaken to assess what would be the most beneficial rice crop management methods for India, comparing SRI with modified, i.e., partially mechanized SRI and with direct-seeded rice, as well as the currently prevailing practices in India: manual transplanting of seedlings with continuous flooding of fields.

## 2. Materials and Methods

Field experiments were conducted for six years, 2012–2013 to 2017–2018, during the wet seasons, June–November, and dry seasons, December to April, at the ICAR-Indian Institute of Rice Research (IIRR), Rajendranagar, Hyderabad, Telangana state (Table S1). The soils at the IIRR research farm (17°33′ N latitude, 78°38′ E longitude) are of medium fertility, with slightly acidic clay loam soil (pH 7.6), low nitrogen (245.9 kg ha$^{-1}$), medium phosphorous (33.9 kg ha$^{-1}$), and medium potassium (184.5 kg ha$^{-1}$) (Table S2).

The first studies compared SRI with MSRI and conventional rice-growing practices to consider the effects of introducing mechanical transplanting together with SRI methods (Experiments 1 and 2). Since the results from SRI and MSRI were quite similar, in the following experiments, MSRI was compared with DSR as well as with CTF (Experiments 3 and 4).

SRI methodology for rice cropping according to its original recommended practices, what can be considered as SRI 1.0 according to Uphoff [14], was assessed in comparison to three other methods of rice crop management: MSRI (mechanically transplanted SRI); DSR (direct-seeded rice using a drum seeder under wet puddled conditions); and CTF (conventional manual transplanting with high plant density and flood irrigation). Trials were replicated three times to minimize the effects of soil differences and measurement errors. The differences in practices are shown in Table 1.

The rice varieties used for the first comparison trials were RP Bio-226 in 2013 and Varadhan in 2013 and 2014, and then RNR-15048 (also known as Telangana Sona) was planted in the trials during the 2015–2018 seasons. These are all high-yielding varieties with a duration of 120–130 days, recommended for being resistant to water stress.

For MSRI trials, a mechanical paddy transplanter powered by a diesel engine VST (V.S. Tiruvengadaswamy Mudaliar tractors and Tillers limited, Bangalore, India) was used to transplant seedlings 16–18 days old from a mat-type nursery. It is able to plant eight rows in a single pass, with a spacing of 24 cm between rows and 10–12 cm between the plants, depending on the speed of the machine.

For the DSR trials, an eight-row drum seeder operated by manual labor modified in the Institute workshop was used to plant germinated seeds in rows spaced 20 cm apart, with 6 cm spacing between plants in each row. Differences among the various methods of crop establishment are represented by the respective plant densities, which were compared in this study, shown in row 3 of Table 1.

**Table 1.** Details of crop establishment methods for rice cultivation.

| Parameters | SRI | MSRI | DSR | CTF |
|---|---|---|---|---|
| Seed rate (kg ha$^{-1}$) | 5 | 12 | 15 | 45 |
| No. of hills m$^{-2}$ | 16 | 42 | 83 | 33 |
| No. of seedlings hill$^{-1}$ | 1 | 3–4 | 2–3 | 3–4 |
| Plant density (m$^{-2}$) | 16 | 125–170 | 165–250 | 100–132 |
| Nursery for raising seedlings | Raised bed, not flooded | Raised bed, mat nursery | No nursery needed | Flooded nursery |
| Nursery (m$^2$ ha$^{-1}$) | 100 m$^2$ | 100 m$^2$ | Nil | 1000 m$^2$ |
| Seedling age at transplanting (days) | 12–14 | 16–18 | Direct sowing by drum seeder in the main field | 30–35 |
| Spacing (cm) | 25 × 25 cm | 24 cm between rows; 10–12 cm between plants | 20 cm between rows; 6 cm between plants | 20 × 15 cm |
| Water management | Alternate wetting & drying (AWD) | AWD method | AWD method | Continuous flooding |
| Weed management | Cono-weeder used 3 times in two directions | Cono-weeder used 3 times in one direction | Cono-weeder used 3 times in one direction | Manual weeding (3 times) |

Weather data including mean minimum and maximum temperatures and precipitation were recorded for wet and dry seasons from 2012–2013 to 2017–2018 (Table 2). Average temperatures during the wet seasons of 2012–2017 ranged from 25.1–26.8 °C. The lowest wet season rainfall (373 mm) was recorded in 2015. Dry season rainfall ranged widely, from an average of 159 mm in 2015 to just 7 mm the following year.

**Table 2.** Averaged weather parameters recorded for years during the period of the experiments.

| Weather Parameters | Wet Season | | | | | |
|---|---|---|---|---|---|---|
| | 2012 | 2013 | 2014 | 2015 | 2016 | 2017 |
| Average temperature (°C) | 25.5 | 25.1 | 26.1 | 26.8 | 25.4 | 24.8 |
| Maximum temperature (°C) | 29.9 | 29.1 | 31.4 | 32.0 | 29.9 | 30.3 |
| Minimum temperature (°C) | 21.0 | 20.7 | 21.8 | 21.6 | 20.8 | 19.4 |
| Total rainfall (mm) | 584.8 | 710.5 | 432.5 | 373.1 | 749.1 | 969.8 |
| | Dry season | | | | | |
| | 2012–2013 | 2013–2014 | 2014–2015 | 2015–2016 | 2016–2017 | 2017–2018 |
| Average temperature (°C) | 25.7 | 24.1 | 23.9 | 26.1 | 24.2 | 26.1 |
| Maximum temperature (°C) | 33.0 | 31.7 | 31.5 | 34.1 | 33.0 | 34.8 |
| Minimum temperature (°C) | 18.4 | 16.4 | 16.3 | 18.1 | 15.3 | 17.5 |
| Total rainfall (mm) | 74.2 | 129.4 | 159.1 | 7.0 | 10.2 | 64.7 |

Grain yield: The rice plants grown in each trial plot (7 × 6 m) were harvested and threshed. Grains from each plot were kept separate and dried under the sun (to 14% moisture), with each plot's grain yield then calculated and recorded in tonnes ha$^{-1}$.

Water productivity: The amount of water applied to each plot was measured using digital water meters, and the total amount supplied to each plot throughout the cropping season was calculated. To maintain the water regime assigned to each plot, elevated bunds

were constructed to separate all plots, and fiber sheets were buried one meter deep around each plot to impede lateral flow.

Water productivity (amount of rice per unit of water) was calculated and expressed in kg ha$^{-1}$ mm$^{-1}$. The frequency of AWD applications for each non-CTF crop establishment method (SRI, MSRI and DSR) was adjusted according to rain events throughout the season. To manage the water supply, the depth of the perched water table in the soil was monitored using a PVC pipe (inner diameter 15 cm; length 40 cm). This was placed into the ground to a depth of 15 cm, and the soil was removed from inside the tube. Water could enter the pipe through perforations in the 15 cm section below the soil's surface, and the farmer could know the amount of water in the soil either by observing the water table directly or by measuring the water depth in the pipe.

For CTF plots, the depth of water in the field was kept at 5 cm up to the rice crop's dough stage, and any extra water was drained from the plot. In all of the treatments, the field's water level was maintained at 2.5 cm depth for the first 10 days following transplanting. After that, the water level in each plot was adjusted according to the treatment prescribed until 10 days before harvest, at which time water was removed from the plots to facilitate harvesting. Water productivity was calculated as the grain yield (kg ha$^{-1}$) divided by the sum of irrigation applications + effective rainfall (in mm).

Fertilization: Recommended doses of NPK (N, P$_2$O$_5$, and K$_2$O at 120, 60 and 40 kg ha$^{-1}$, respectively) were applied to all plots since variations in exogenous fertilization of soil were not a factor being evaluated. Nutrient applications were thus the same for all of the trials. Nitrogen was supplied 50% from inorganic and 50% from organic sources, the latter being farmyard manure (FYM). The urea was provided in 3 splits (1/2 as basal application and then $\frac{1}{4}$ at both 30 DAT and 50 DAT); single super-phosphate was applied basally at planting; and muriate of potash was given in 2 splits (a basal application and @ 50 DAS). In principle, the fertilization of the SRI plots should have been entirely organic; however, sources and amounts of externally supplied nutrients were not a variable in these trials.

Energy use efficiency: Energy indices derived from previously published studies were used to calculate the energy equivalence of inputs and outputs for the respective crop establishment methods. Inputs included human labor, machinery, farmyard manure (FYM), chemical fertilizers, plant protection chemicals, herbicides, and electricity. The tasks of weeding, watering, and application of FYM, fertilizer, and pesticides were carried out by human labor, while harvesting and the preparation of land were accomplished with machines.

Paddy grain and paddy straw were considered as the outputs. Calculations were made to determine the energy represented by these outputs to compare them with the energy embodied in the respective inputs per hectare. To estimate energy efficiency, the input and output values were the corresponding energy-equivalence coefficients shown in Annexure III [15–23].

The total energy needed for labor, farm equipment, seed, fertilizer, and irrigation consumption in each system was added up, and the associated output was also summarized in terms of energy, denominated in GJ ha$^{-1}$. The energy output represented by the main product (grain) and by-product (straw) was summarized by multiplying the production by their respective energy equivalents. Energy use efficiency (EUE) was computed as the gross energy output (in GJ ha$^{-1}$) × 100, divided by total energy input (in GJ ha$^{-1}$).

GHG estimation: The closed-chamber method was used to assess plant-mediated CH$_4$ and N$_2$O emissions from the experimental plots at weekly intervals during the entire growing season. Samples were taken using chambers (50 cm × 30 cm × 100 cm) built of 6 mm acrylic sheets kept over aluminum stands inserted in the soil (Figure 1). To make the system airtight, channels at the base of the aluminum stands were filled with water.

Samples were taken into 20 mL polypropylene syringes with a three-way stopper using a hypodermic needle (24 gauge) through a rubber septum at the top of the chamber. A thermometer was placed into the chamber through a different septum to measure the

temperature during the sampling time. A small DC-driven fan powered by a 9-volt battery was used to homogenize the air inside the chamber.

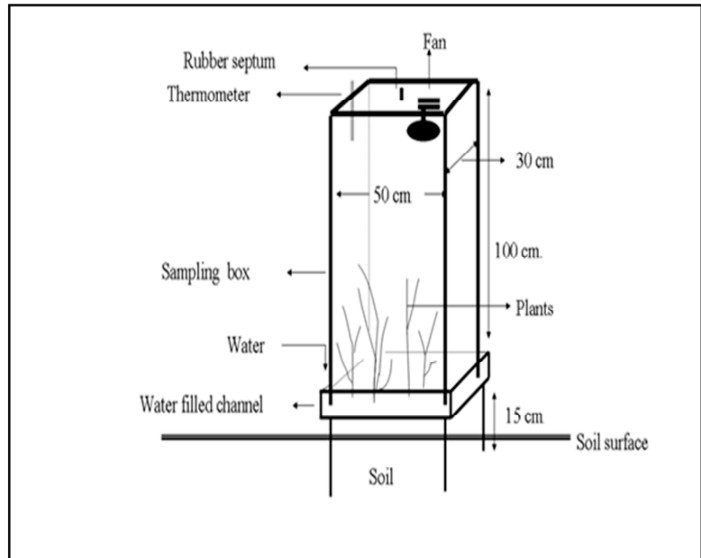

**Figure 1.** Gas sample collection was performed using a closed-chamber technique.

Data on air temperature and precipitation were gathered from the farm's weather station. Gas samples of approximately 20 mL each were collected on the day of sampling at 0, half hour, and one hour. The same sampling schedule was followed in all the seasons studied. A gas chromatograph (Model 450-GC, Varian Inc., Walnut Creek, CA, USA) equipped with an electron capture detector, a flame ionization detector, and a thermal conductivity detector were used to evaluate the gas samples obtained for their $N_2O$ and $CH_4$ immediately.

Global warming potential (GWP) is an index that summarizes the effects of the respective GHGs that trap heat in the atmosphere, thereby contributing to global warming. Their relative and total contributions to global warming potential (GWP) are by convention evaluated in relation to a single gas, $CO_2$. The GWP for $CH_4$ (based on a 100-year time horizon) is 25, and for $N_2O$, it is 298, when the GWP for $CO_2$ is considered as having a value of 1. In this study, $CO_2$ emissions were not considered because they are not much affected by management methods. GWP was calculated using the equation: GWP = [kg $CH_4$ × 25] + [kg $N_2O$ × 298] [24].

Greenhouse gas intensity: An index of greenhouse gas intensity (GHGI) summarizes the level of emissions per unit of grain yield produced, indicating the amount of emissions released per unit production of grain. It was calculated by dividing global warming potential (GWP) by grain yield [25].

$$\text{GHGI (kg } CO_2 \text{ eq kg}^{-1} \text{ grain)} = \text{GWP (kg } CO_2 \text{ eq ha}^{-1})/\text{Grain yield (kg ha}^{-1})$$

Soil microbial population and enzyme activities: In this study, the pre-treatment microbiological parameters were considered to be in a steady state, with subsequent differences in soil microbial populations and enzyme activities regarded as the result of the respective management practices. Soils from the rhizosphere of rice plants grown under the respective crop establishment methods were sampled to determine their microbial populations and enzyme activities.

Serial dilution and agar plating methods were used for the enumeration of populations of bacteria, fungi, and actinomycetes. Nutrient agar medium [26] was used for the enumeration of total heterotrophic bacteria. Fungi populations were estimated on Martin's rose Bengal agar medium, containing 1.25 g of streptomycin and 0.033 g of rose

Bengal in 1 L of the medium [27]. Actinomycetes populations were enumerated using Kuster's agar medium [28]. Fluorescein diacetate (FDA) hydrolytic activity in the soils was estimated using fluorescein diacetate as the hydrolysis substrate [29]. Key soil enzymes, i.e., dehydrogenase, β glucosidase, alkaline phosphatase, and arylsulfatase, were determined by using standard methods [30].

Soil nematode communities: To provide a baseline understanding of the soil nematode community, soil samples were collected at a depth of 20 cm with a shovel from both the SRI and CTF trial plots before the trials were started. Eighteen replicate plots (5 m$^2$) were maintained for each system. The initial soil samples taken from both the SRI and CTF experimental plots showed no significant differences in the composition of the soil nematode communities.

The rice root nematode (*Hirschmanniella* spp.) comprised more than 60% of all the plant-parasitic nematodes identified, and they dominated the plant-parasitic nematode communities in both SRI and CTF plots. Other minor ectoparasitic nematodes included *Helictylenchus* spp. and *Psilenchus* spp. More harmful species like *Meloidogyne graminicola*, which causes root-knot disease, and the *Pratylenchus* species, which cause root lesions, were absent in the samples.

Long-term changes in the composition of soil nematode communities were assessed by analyzing the soil samples collected from the field plots after six years, i.e., after the completion of twelve crop cycles: six wet and six dry seasons. Nematode extraction was performed using modified Cobb's sieving and decanting technique with 100 g sub-samples taken from each composite soil sample. Nematode enumeration and identification of nematode trophic groups were performed using diagnostic keys [31]. The total number of plant-parasitic and free-living microbial-feeding nematodes in each sample was counted by observing nematode suspension under an inverted microscope at 40× magnification.

Economic analysis: The costs of cultivation were recorded for inputs such as seed, manures, fertilizers, irrigation, and plant protection chemicals listed, using current market prices, and then summed up. Similarly, the expenditures incurred for the operations involved in cultivation, such as tillage/land preparation, nursery-raising, transplanting, harvesting and threshing, were added up, and the costs of hiring tractor-driven machinery and the wages of human laborers (based on eight hours of work per day) were included in the cost of cultivation. The Government of India's minimum support price (MSP) for rice was used to calculate the value of production [32] and then to calculate the gross return. The net return was determined according to the equation:

$$\text{Net return} = \text{Gross return} - \text{Cost of cultivation}$$

A summary analysis was carried out for the economic returns from each crop establishment method as this is what farmers and policy-makers are most concerned about. A benefit–cost ratio (BCR) is the ratio between the economic return from production and the cost of that production, both summarized in terms of Rs. ha$^{-1}$, reported as a ratio.

Statistical analysis: Before performing the analysis of variance, descriptive statistics were calculated for the study variables across the different crop establishment methods. In the following sections, the data analysis was carried out with a two-factorial randomized block design, considering the method of crop establishment as one factor and crop season as another factor, using SAS version 9.3 [33,34] available at the ICAR-Indian Institute of Rice Research in Hyderabad.

## 3. Results

Grain yield: The basic System of Rice Intensification (SRI) methods resulted in significantly higher grain yields (6.23–6.47 t ha$^{-1}$), about 18% more than partially mechanized SRI (4.75–5.72 t ha$^{-1}$) (Table 3, Experiments 1 and 2). In turn, the MSRI method was found to give a significantly higher yield (6.27–6.41 t ha$^{-1}$) than both DSR (6.02–6.09 t ha$^{-1}$) and conventional transplanting (5.36–5.59 t ha$^{-1}$) (Table 3, Experiment 3). The mean yields

from SRI, DSR and MSRI were significantly higher than CTF in all experiments as seen in Table 3.

**Table 3.** Yield performance (t ha$^{-1}$) of crop establishment methods across the four experiments.

| Method of Establishment | Grain Yield (t ha$^{-1}$) | | | | | | |
|---|---|---|---|---|---|---|---|
| | Experiment 1 | | Experiment 2 | | Experiment 3 | | Experiment 4 |
| | Wet Season | Dry Season | Wet Season | Dry Season | Wet Season | Dry Season | Wet Season |
| SRI | 6.23 [a] | 6.47 [a] | 6.09 [a] | 6.23 [a] | - | - | - |
| MSRI | 4.75 [b] | 5.02 [b] | 5.72 [b] | 5.65 [b] | 6.27 [a] | 6.41 [a] | 5.07 [a] |
| DSR | - | - | - | - | 6.02 [a] | 6.09 [a] | - |
| CTF | 4.10 [c] | 4.44 [c] | - | - | 5.59 [b] | 5.36 [b] | 4.64 [b] |
| SEm± | 0.05 | 0.01 | 0.03 | 0.04 | 0.06 | 0.10 | 0.32 |
| CD ($p \leq 0.05$) | 0.28 | 0.15 | 0.24 | 0.28 | 0.33 | 0.41 | 0.09 |
| $p \geq$ F | 0.07 | 0.0002 | 0.36 | 0.07 | 0.11 | 0.54 | 0.23 |

SEm = Standard error of the mean, CD = Critical difference. Means within the same column followed by the same small letter are not statistically different (CD $p < 0.05$).

Water productivity: This was calculated in kg of paddy rice harvested per hectare-millimeter of water (per 10,000 L). As shown in Table 4A, this productivity for SRI was 5.32–6.85 kg ha-mm$^{-1}$, and for MSRI 4.14–5.72 kg ha-mm$^{-1}$, followed by DSR with 5.06–5.11 kg ha-mm$^{-1}$, all greater than 3.52–4.56 kg ha-mm$^{-1}$ for CTF. All three methods (SRI, MSRI and DSR) were significantly superior to CTF in terms of their water productivity irrespective of the season.

Economic productivity: As seen in Table 4B, the benefit–cost ratios calculated from our trials showed some season-to-season variation, due mostly to differences in climate and resulting yield. However, the benefit/cost ratio for SRI was consistently greater than that for the other management methods (Experiments 1 and 2), and about 10% higher than for MSRI, one of the comparisons of most interest for this study. The results from Experiment 3 showed the B:C ratio for MSRI to be 18% greater than DSR and 30% more than CTF.

**Table 4.** Water productivity (kg ha-mm$^{-1}$) and B:C ratios of the crop establishment methods across four experiments.

| Method of Establishment | A. Water productivity (kg ha-mm$^{-1}$) | | | | | | |
|---|---|---|---|---|---|---|---|
| | Experiment 1 | | Experiment 2 | | Experiment 3 | | Experiment 4 |
| | Wet Season | Dry Season | Wet Season | Dry Season | Wet Season | Dry Season | Wet Season |
| SRI | 5.53 [a] | 6.83 [a] | 5.32 [a] | 5.32 [a] | - | - | - |
| MSRI | 4.14 [b] | 5.12 [b] | 5.16 [b] | 5.16 [b] | 5.48 [a] | 5.67 [a] | 5.72 [a] |
| DSR | - | - | - | - | 5.06 [b] | 5.11 [b] | - |
| CTF | 3.52 [c] | 4.50 [c] | - | - | 4.42 [c] | 4.56 [c] | 4.18 [b] |
| SEm± | 0.05 | 0.08 | 0.003 | 0.01 | 0.02 | 0.02 | 0.13 |
| CD ($p \leq 0.05$) | 0.3 | 0.37 | 0.08 | 0.08 | 0.16 | 0.17 | 0.46 |
| $p \geq$ F | 0.10 | 0.05 | 0.0008 | 0.08 | 0.05 | 0.41 | 0.0003 |

**Table 4.** *Cont.*

| Method of Establishment | B. Economic Productivity (Benefit/Cost Ratio) | | | | | | |
|---|---|---|---|---|---|---|---|
| | Experiment 1 | | Experiment 2 | | Experiment 3 | | Experiment 4 |
| | Wet Season | Dry Season | Wet Season | Dry Season | Wet Season | Dry Season | Wet Season |
| SRI | 3.12 [a] | 2.93 [a] | 1.42 [a] | 1.44 [a] | - | - | - |
| MSRI | 2.69 [b] | 2.67 [b] | 1.34 [b] | 1.31 [b] | 1.48 [a] | 1.52 [a] | 1.91 [a] |
| DSR | - | - | - | - | 1.33 [b] | 1.21 [b] | - |
| CTF | 2.21 [c] | 2.14 [c] | - | - | 1.15 [c] | 1.16 [c] | 1.63 [b] |
| SEm± | 0.02 | 0.02 | 0.004 | 0.001 | 0.01 | 0.01 | 0.13 |
| CD ($p \leq 0.05$) | 0.17 | 0.2 | 0.03 | 0.05 | 0.13 | 0.15 | 0.04 |
| $p \geq$ F | 0.002 | 0.003 | 0.02 | 0.01 | 0.02 | 0.89 | 0.45 |

SEm = Standard error of the mean, CD = Critical difference. Means within the same column followed by the same small letter are not statistically different (CD $p < 0.05$).

Energy use efficiency: In experiments 1 and 2, SRI methods showed 11.8% higher output of energy compared to the energy inputs made. This was 22% more than MSRI methods, which showed energy use efficiency of 9.7%. This reflected SRI's greater biomass production and grain yield. In experiment 3, DSR methods also showed slightly higher (4%) mean energy efficiency than MSRI, 10.1% compared to 9.7%. In terms of energy efficiency, all three methods (SRI, MSRI, and DSR) were significantly superior to CTF (8.3%) (Table 5).

**Table 5.** Effects of crop establishment methods on energy use efficiency.

| Method of Establishment | Energy Use Efficiency (%) | | | | | | | | | |
|---|---|---|---|---|---|---|---|---|---|---|
| | Experiment 1 | | Experiment 2 | | Experiment 3 | | Experiment 4 | Wet Season Mean | Dry Season Mean | Total Mean |
| | Wet Season | Dry Season | Wet Season | Dry Season | Wet Season | Dry Season | Wet Season | | | |
| SRI | 11.44 [a] | 11.76 [a] | 12.30 [a] | 11.79 [a] | - | - | - | 11.9 | 11.8 | 11.8 |
| MSRI | 8.94 [b] | 9.25 [b] | 10.55 [b] | 10.49 [b] | 10.45 [a] | 10.84 [a] | 7.09 [a] | 9.7 | 10.2 | 9.7 |
| DSR | - | - | - | - | 10.08 [b] | 10.15 [b] | - | 10.1 | 10.2 | 10.1 |
| CTF | 7.02 [c] | 7.48 [c] | - | - | 9.82 [c] | 10.08 [c] | 6.34 [b] | 7.7 | 8.8 | 8.3 |
| SEm± | 0.12 | 0.04 | 0.002 | 0.003 | 0.02 | 0.04 | 0.1 | | | |
| CD ($p \leq 0.05$) | 0.44 | 0.27 | 0.02 | 0.02 | 0.20 | 0.25 | 0.33 | | | |
| $p \geq$ F | 0.004 | 0.01 | 0.0001 | 0.0001 | 0.001 | 0.86 | 0.15 | | | |

SEm = Standard error of the mean; CD = Critical difference. Means within the same column followed by the same small letter are not statistically different (CD $p < 0.05$).

Greenhouse gas (GHG) emissions: Crop establishment methods significantly affected methane and nitrous oxide emissions. The highest seasonal emission of methane ($CH_4$) was observed with the conventional CTF methods (26.9 to 36.6 kg ha$^{-1}$ season$^{-1}$ in Experiments 1 and 3), while the lowest was with SRI methods (18.9–21.6 kg ha$^{-1}$), which was one-third less. $CH_4$ emissions were even lower for MSRI (11.6 to 20.6 kg ha$^{-1}$ season$^{-1}$ (Experiment 3) in comparison to CTF (27.8 to 36.6 kg ha$^{-1}$ season$^{-1}$). $CH_4$ emissions from the different methods followed the order of CTF > DSR > MSRI > SRI, indicating the greater impact SRI methods in this regard (Table 6). Of particular interest, in the wet season, $CH_4$ emissions with SRI and MSRI methods were relatively less compared to the other rice-establishment methods. Nitrous oxide fluxes, on the other hand, were relatively lower from CTF (6.5 to 10.1 kg ha$^{-1}$ season$^{-1}$) in Experiments 1 and 3 than from MSRI and SRI (7.3 to 10.7, and 10.1 to 10.3 kg ha$^{-1}$ season$^{-1}$, respectively). The difference is due to the respective soil conditions being more aerobic or more hypoxic.

**Table 6.** Effect of crop establishment methods on GHG emissions.

| Method of Establishment | A. Methane ($CH_4$) Emissions (kg ha$^{-1}$ Season$^{-1}$) | | | | | | | |
| --- | --- | --- | --- | --- | --- | --- | --- | --- |
| | Experiment 1 | | | | Experiment 3 | | Wet Season Mean | Dry Season Mean |
| | Wet Season 2012 | Dry Season 2012–2013 | Wet Season 2013 | Dry Season 2013–2014 | Wet Season 2015 | Wet Season 2016 | | |
| SRI | 20.6 [b] | 18.9 [c] | 21.6 [b] | 20.9 [b] | - | - | 21.1 | 19.9 |
| MSRI | 25.0 [a] | 22.1 [b] | 21.6 [b] | 23.9 [b] | 11.6 [b] | 20.6 [c] | 19.7 | 22.9 |
| DSR | - | - | - | - | 26.0 [a] | 32.4 [b] | 29.2 | - |
| CTF | 26.9 [a] | 27.1 [a] | 29.6 [a] | 28.3 [a] | 27.8 [a] | 36.6 [a] | 30.2 | 27.7 |
| SEm± | 0.71 | 0.74 | 0.8 | 1.04 | 0.85 | 0.74 | | |
| CD ($p \leq 0.05$) | 2.77 | 2.92 | 3.13 | 4.08 | 3.98 | 2.92 | | |
| Probability of significance | 0.0008 | <0.0001 | <0.0001 | 0.0001 | <0.0001 | <0.0001 | | |

| Method of Establishment | B. Nitrous Oxide ($N_2O$) Emissions (kg ha$^{-1}$ Season$^{-1}$) | | | | | | | |
| --- | --- | --- | --- | --- | --- | --- | --- | --- |
| | Experiment 1 | | | | Experiment 3 | | Wet Season Mean | Dry Season Mean |
| | Wet Season 2012 | Dry Season 2012–2013 | Wet Season 2013 | Dry Season 2013–2014 | Wet Season 2015 | Wet Season 2016 | | |
| SRI | 10.1 | 10.3 | 10.3 | 10.3 | | | 10.2 | 10.3 |
| MSRI | 10.2 | 10.5 | 10.5 | 10.6 | 10.7 | 7.3 | 9.7 | 10.5 |
| DSR | - | - | - | - | 10.2 | 7.3 | 8.8 | - |
| CTF | 9.9 | 10.0 | 10.1 | 10.1 | 10.1 | 6.5 | 9.2 | 10.0 |
| SEm± | 0.36 | 0.41 | 0.41 | 0.22 | 0.36 | 0.28 | | |
| CD ($p \leq 0.05$) | NS | NS | NS | NS | NS | 0.33 | | |
| Probability of significance | 0.72 | 0.41 | 0.42 | 0.52 | 0.083 | 0.0004 | | |

**Table 6.** *Cont.*

| Method of Establishment | C. Global Warming Potential (GWP) (kg $CO_2$-eq $ha^{-1}$) | | | | | | | | | |
| | Experiment 1 | | | | Experiment 3 | | Wet Season Mean | Dry Season Mean | GHG $kg^{-1}$ Grain Wet Season | GHG $kg^{-1}$ Grain Dry Season |
| | Wet Season 2012 | Dry Season 2012–2013 | Wet Season 2013 | Dry Season 2013–2014 | Wet Season 2015 | Wet Season 2016 | | | | |
| SRI | 3512 | 3552 | 3619 | 3602 | - | - | 3565 | 3577 | 0.58 | 0.56 |
| MSRI | 3671 | 3680 | 3710 | 3742 | 3488 [b] | 2692 [b] | 3390 | 3711 | 0.62 | 0.65 |
| DSR | - | - | - | - | 3705 [a] | 2986 [a] | 3346 | - | 0.56 | - |
| CTF | 3635 | 3657 | 3735 | 3705 | 3720 [a] | 2861 [a] | 3488 | 3681 | 0.73 | 0.75 |
| SEm± | 107.8 | 112.9 | 130.8 | 86.2 | 43 | 39 | | | | |
| CD ($p \leq 0.05$) | NS | NS | NS | NS | 170 | 154 | | | | |
| Probability of significance | 0.24 | 0.12 | 0.34 | 0.29 | 0.008 | 0.0004 | | | | |

SEm = Standard error of the mean, CD = Critical difference. Means within the same column followed by the same small letter are not statistically different (CD $p < 0.05$). NS = Not Significant.

Methane and nitrous oxide respond differently to irrigation practice, i.e., flooding vs. AWD. $CH_4$ is reduced by changing from continuous flooding more aerobic soil conditions, while this change increases $N_2O$. So, what is important is the net effect on global warming potential (GWP) resulting from the different management systems. This requires a summation of effects, denominating any increases or decreases in $CH_4$ or $N_2O$ in terms of a common metric, in this case, their equivalence to carbon dioxide ($CO_2$).

The differences among the four systems in terms of their impact on GWP $ha^{-1}$ during the wet and dry seasons were neither great nor significant as seen in Table 6. The results in these trials were not consistent with what has been reported from other evaluations carried out in India and elsewhere, which have shown net reductions in GWP $ha^{-1}$ when SRI methods were used compared to conventional rice cultivation. These have shown net reductions in GWP $ha^{-1}$ when SRI methods were used compared to conventional methods of rice cultivation [35–40].

Microbial populations and enzyme activities: SRI methods supported significantly higher bacterial, fungal, and actinomycetes populations (respectively, 7.20, 5.22, and 4.62 log CFU $g^{-1}$ soil) as compared to conventional transplantation and flooding (6.7, 4.66 and 3.86 log CFU $g^{-1}$ soil, respectively). It was observed that under SRI methods of cultivation, the bacterial population increased by 8%, the fungal population by 12%, and the actinomycetes by 20% compared with CTF.

Also, we note that with SRI, two commonly used parameters of soil quality—soil dehydrogenase and FDA activity—were significantly higher than with CTF, respectively, by 8.5% and 15.8%. Also, a large increase (78%) was observed in glucosidase activity (91.2 µg p-nitrophenol $g^{-1}$ soil $h^{-1}$) in SRI soil relative to CTF (51.2 $g^{-1}$ soil $h^{-1}$). SRI plots also had higher but not significantly higher activities of other enzymes like alkaline phosphatase and arylsulfatase over CTF plots (Table 7).

**Table 7.** Soil microbial populations and enzyme activities under different crop establishment methods.

| Parameters | SRI | CTF |
|---|---|---|
| Microbial populations (log CFU $g^{-1}$ dry soil) * | | |
| Bacteria | 7.20 [a] | 6.67 [b] |
| Fungi | 5.22 [a] | 4.66 [b] |
| Actinomycetes | 4.62 [a] | 3.86 [b] |
| Soil enzyme activities | | |
| Dehydrogenase (µg TPF $g^{-1}$ soil 24 $h^{-1}$) | 196.08 [a] | 180.73 [b] |
| Fluorescein diacetate hydrolytic activity (µg $g^{-1}$ dry soil 0.5 $h^{-1}$) | 51.06 [a] | 44.08 [b] |
| Glucosidase activity (µg p-nitrophenol $g^{-1}$ soil $h^{-1}$) | 91.24 [a] | 51.18 [b] |
| Phosphatase activity (mg p-nitrophenol $g^{-1}$ soil $h^{-1}$) | 1.23 [a] | 1.18 [a] |
| Arylsulfatase activity (mg p-nitrophenol $g^{-1}$ soil $h^{-1}$) | 7.61 [a] | 7.35 [a] |

* Means are pooled data from dry seasons 2015 and 2016). Means within the same row followed by the same small letter are not statistically different. (CD $p \leq 0.05$).

Effects on soil nematode community: Nematode analyses on nematode abundance than CTF (Figure 2). While there were more nematodes in total associated with soil under SRI management, the numbers of plant-parasitic nematodes (PPN) were less than with CTF, and the SRI plots had a substantially greater abundance of microbial-feeding nematodes than in CTF soil.

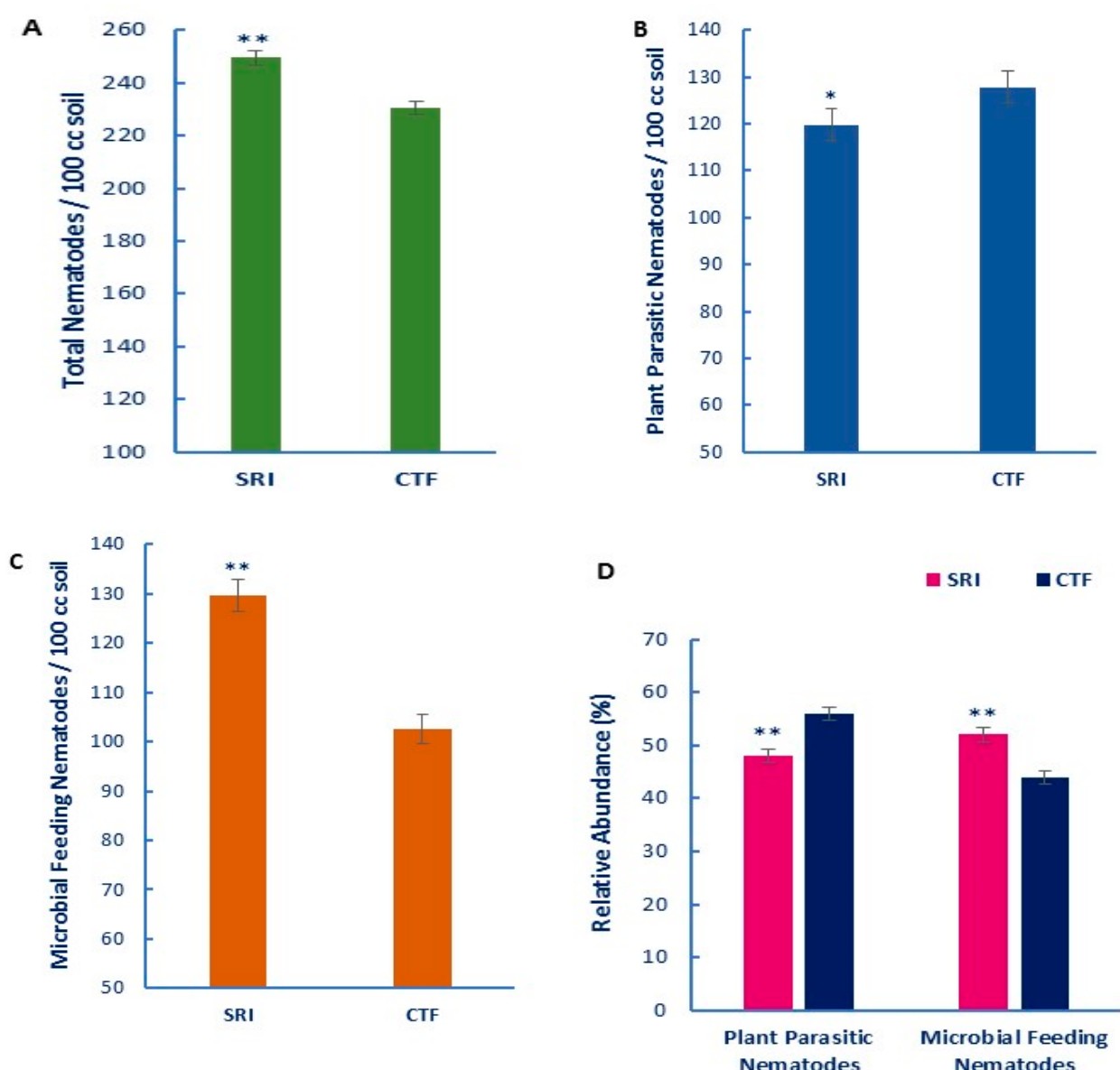

**Figure 2.** Abundance (mean ± SE) of (**A**) total nematodes, (**B**) plant-parasitic nematodes, (**C**) microbial-feeding nematodes, and (**D**) the relative abundance of plant-parasitic vs. microbial-feeding nematodes in rice plots maintained under System of Rice Intensification (SRI) vs. the conventional transplanting and flooding (CTF). Bars with stars indicate significant differences (** $p = < 0.01$; * $p = < 0.05$).

Under SRI management, the relative abundance of nematodes that feed on microorganisms and are thus benign, even beneficial, for plants, was 52%, compared with 45% for CTF. Conversely, with SRI, the relative abundance of nematodes that parasitize plants was lower, 48%, as compared to 55% with CTF. This could be one of the effects that are contributing to SRI's higher yield.

## 4. Discussion

In this study making various comparisons over six years of trials, the use of SRI methods as originally recommended, with manual transplanting, yielded the highest economic returns compared to the other three methods evaluated. This confirmed that crop yield can be increased by reducing plant density m$^{-2}$, with all of the plants using the available resources of water, nutrients, and sunlight as fully as possible [41]. Further, the

best results from rice crops established by transplanting are attained when seedlings are transplanted before the start of their 4th phyllochron of growth having optimum square spacing, usually ~16 hills m$^{-2}$, to facilitate rooting and tillering [42,43].

Experience with the irrigation method of alternate wetting and drying indicates that yield is not reduced if rice fields are reflooded when their water level (FWL) reaches ~15 cm below the soil surface [44]. Our results were consistent with this as we found no significant yield penalty when this kind of 'safe' AWD was practiced with any of the three methods of crop establishment (SRI, MSRI and DSR), compared to CTF [45,46].

A greater economic benefit from both MSRI and SRI was seen from their B:C ratios in these trials. MSRI has a particular attraction because of its labor-saving, which lowers the costs of production. Thus, despite the several advantages of manual SRI over MSRI seen in our trial results, given the labor constraints faced by many, even most rice-producers, there is reason to modify SRI practices with mechanization wherever labor is a constraint or costly.

The availability of suitable implements for the mechanized transplanting of young seedlings, optimally spaced, would promote SRI uptake on a larger scale in India, particularly in areas where agricultural labor is scarce. Experimentation with improved designs for mechanical transplanters is ongoing in India, and it can be anticipated that there will soon be more appropriate implements for mechanical transplanting than were available for our trials.

Engineers at Tamil Nadu Agricultural University have developed a four-row transplanter shown in Figure 3 that can place single young seedlings (although sometimes more) in a square grid pattern. Results from on-farm trials with 50 farmers, 10 in each of five villages in Kancheepuram district of Tamil Nadu state, have been reported [47]. These trials compared manually transplanted SRI with mechanical transplanting (MSRI) as well as with conventional hand transplanting and usual farmer practices.

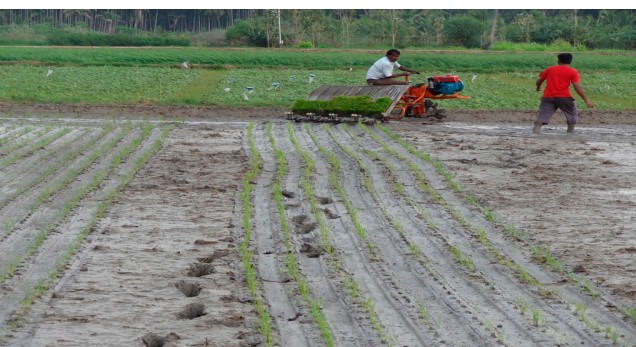

**Figure 3.** Mechanical transplanter designed at Tamil Nadu Agricultural University for SRI use, transplanting young mat-nursery seedlings, 1–2 hill$^{-1}$ with a spacing of 23 × 23 cm that permits mechanical weeding of the field in perpendicular directions.

The TNAU evaluation indicated that aboveground plant growth was superior with the original SRI methods, but that root growth was more with MSRI, for reasons not evident. In these trials, SRI yield ha$^{-1}$ was 2% higher than with MSRI, not a significant difference; the 20% higher yield with SRI methods compared to CTF was significant. Costs of production ha$^{-1}$ with SRI were 7% lower compared to conventional methods, so net income ha$^{-1}$ was 38% greater; with MSRI, the net income was 35% more. This difference reflected the performance of the mechanical transplanter used in the trials, however, and improved designs of mechanical transplanters could give results more favorable to MSRI.

In our trials, because of constraints in machine design, the version of MSRI that was evaluated did not provide this much reduction in plant density, so some of the benefits of SRI's proposed practices were probably forgone. If and when a more suitable mechanical transplanter is designed and used, it is anticipated that MSRI will become more advantageous economically, with a higher yield and reduced labor requirements.

For SRI to expand in India, there is also a need to develop appropriate motorized implements for mechanical weeding of SRI rice crops. These implements would cover multiple rows simultaneously and utilize mechanical power instead of human energy to propel the weeder. Development and timely availability of durable, effective, and affordable implements for weeding will make SRI adoption more attractive to farmers, helping them to capitalize more efficiently upon the biological processes and potentials that SRI methods tap.

In our trials, the energy input of labor was significantly reduced with SRI, by 36%. This was due in part to the reduced labor needed to manage much smaller, short-duration nurseries (8 to 12 days) [48] as compared to conventional nurseries that are $10\times$ larger and need tending for 25 days or more. Also, less labor was required for transplanting $10\times$ fewer seedlings in SRI as compared to the conventional method [49,50].

The AWD method of irrigation followed in our SRI, MSRI and DSR trials recorded relatively lower emissions of methane due in large part to the more aerobic soil conditions resulting from intermittent drying and wetting (Table 6). SRI trials showed 30% lower $CH_4$ emissions compared to CTF, and MSRI methane emissions were 27% lower. The DSR emissions were on par with CTF.

Regarding nitrous oxide ($N_2O$), TPF not surprisingly recorded the lowest emissions of $N_2O$-N. However, the other three crop establishment methods showed little difference in this respect. The slightly higher $N_2O$ emissions with SRI (7% more than CTF) did not offset and cancel out SRI's greater reductions in $CH_4$. We note that the weightings for calculating GWP assume a 100-year time horizon. Within a shorter, 20-year time horizon, $CH_4$ is 80 times more potent than carbon dioxide ($CO_2$), so methane is the greenhouse gas that needs to be most urgently curtailed.

The most effective way to reduce greenhouse gas emissions from paddy fields is to reduce their $CH_4$ emission through changes in water management [51]. This makes replacing continuous flooding with AWD a priority policy concern, especially because this reduces the demand for water, an increasingly scarce resource, and can also raise yield.

The use of inorganic N fertilizers is also a significant contributor to GHG emissions. This study did not evaluate alternative sources of fertilization, however, so it could not ascertain how much GHG reduction is possible by moving rice production away from its current reliance on synthetic fertilizers and toward organic sources of nutrients for soil organisms and plants.

A study of SRI effects in Telangana state by researchers from Oxford University and India's National Institute for Rural Development assessed GHG emissions and other effects. This analysis found that smallholders using SRI methods in addition to increasing their yield by 60% and reducing their use of groundwater and fossil fuel, respectively, by 60% and 74%, decreased their GHG emissions by 40% ha$^{-1}$ and by 60% kg$^{-1}$ of rice produced. Their evaluation was more inclusive than we were able to perform because it also considered $CO_2$ emissions throughout the whole process of rice production, including the manufacture of inputs [52].

Our results showed, as anticipated, that SRI practices enhance the soil food web by providing a more favorable environment for beneficial soil organisms, ranging from microbes to earthworms. They documented a build-up of beneficial microbe-feeding nematodes that contribute to processes like decomposition and nutrient mineralization within the rice ecosystem, which have positive effects on crop growth and productivity.

The yield gains with SRI management need not be compromised by an increase in the total nematode population under aerobic soil conditions. In our six year trials, the nematode community under SRI was dominated by less-pathogenic species. This may not be the case in fields that have endemic populations of more pathogenic species, like the root-knot nematode *Meloidogyne graminicola*. However, researchers in Thailand have reported that rice yields were lower with SRI than with CTF due to a rapid build-up of rice root-knot nematodes under SRI water management. Farmers should therefore, be cautioned when adopting SRI to pay attention to the status of parasitic nematodes species.

Microbial populations under the SRI method of crop establishment are increased by greater root exudation, by having more organic matter in the soil due to weed incorporation, and by more aerobic soil conditions compared to conventional inundated rice cultivation [53]. In our study, increases were observed in dehydrogenase enzyme activity representing microbial oxide reduction processes; in fluorescein diacetate hydrolysis, which indicates the presence of enzymes like lipases, esterases, and proteases; and in glucosidase, which has a critical role in carbon cycling. These effects could be attributed to the enhanced abundance and activity of soil microbes resulting in particular from having more root exudation from larger root systems. There would also be greater carbon mineralization from increasing organic inputs into the soil with SRI methods, but this was not evaluated in our trials.

In this study, enhanced levels of other beneficial enzymes like alkaline phosphatase and arylsulfatase were also observed with SRI methods of cultivation [54,55]. The more aerobic soil conditions sustained with SRI, resulting from reduced irrigation and the use of a cono-weeder to control weeds, disturbing the surface soil around the rice plants several times, create a more favorable environment for the soil microflora and its activities than results from conventional rice cultivation methods.

## 5. Conclusions

This multi-year study found SRI methods of production helping to achieve higher yields while lowering farmers' costs for production, making this mode of cultivation more profitable. SRI methodology uses less water and fewer agrochemicals while generating significantly more income. The modification of SRI by mechanizing transplanting (MSRI) produced results that were mostly on par with SRI, and these results could probably be improved by having a transplanter specifically designed to accommodate younger seedlings (<15 days) and to regulate the spacing of plants at both directions with one or at most two seedlings per hill.

Similarly, direct seeding could be made more productive by better approximating the spacing of manually transplanted SRI. Direct seeding is gaining farmer acceptance in parts of India and elsewhere to replace the transplanting of rice, but farmers need to be very careful to manage the weeds with suitable and comprehensive weed management practices. The DSR evaluated in this study could be improved upon for SRI purposes.

To support the adoption of SRI more generally, the mechanization for land levelling, crop establishment, and mechanical weeding should be promoted to make SRI more attractive to farmers. The availability of appropriate implements would also make SRI more adaptable for large-scale operations.

The overall conclusion from these six years of evaluation is that current rice production methods are no longer desirable—manual transplanting and flooding of paddies, using older seedlings and relying more on inorganic than on organic sources of soil and plant nutrition. SRI ideas and methods can make extensive improvements in the rice sector in India as well as elsewhere. From its inception, SRI has been described as a work in progress, and that designation still applies.

**Supplementary Materials:** The following supporting information can be downloaded at: https://www.mdpi.com/article/10.3390/agronomy13102492/s1, Table S1: Experiment details; Table S2: Physico-chemical properties of the experimental soil.

**Author Contributions:** R.M.K., R.M.S., P.C., N.S., P.C.L., P.B.B.B.M., S.K., J.V.N.S.P., T.V.S., M.D.T. and B.S. (Banugu Sreedevi)—Conceptualization, methodology, investigation, validation, data curation, analysis, writing—original draft preparation, writing—review and editing, visualization, supervision; D.S., N.B., M.N.A., S.R. and B.S. (Banda Sailaja)—Formal analysis, S.P.M., D.S., M.D.T. and S.V.—literature search, data curation, input for manuscript. All authors have read and agreed to the published version of the manuscript.

**Funding:** This research received no external funding.

**Institutional Review Board Statement:** Article approved for submission; Ref No. IIRR/PD/DIR/PMEC/2023-24 Res.paper/484/dt. 12-05-2023.

**Informed Consent Statement:** Consent taken from all the concerned.

**Data Availability Statement:** The data presented in this study are available on request from the corresponding author.

**Acknowledgments:** Authors would like to thank the ICAR-Indian Institute of Rice Research for providing facilities for the conduct of the study. The authors would also like to thank Norman Uphoff, Cornell University, USA and Gururaj Katti, ICAR-IIRR Crop Protection Section, for providing their critical inputs for the preparation of this manuscript, and all of the students and research scholars who were associated with these multi-year trials.

**Conflicts of Interest:** The authors declare no conflict of interest.

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
