# Peer review of "Comparison of System of Rice Intensification Applications and Alternatives in India: Agronomic, Economic, Environmental, Energy, and Other Effects"

_agronomy, doi:10.3390/agronomy13102492_

Round 1

Reviewer 1 Report (New Reviewer)

Recommendations to authors

General comments:

- The article has an interesting topic as it compares intensification methods for rice, a crop of great importance for India and worldwide. However, the topic is not new, and the study lacks a clear presentation of its unique contributions compared to the existing literature, so at least the discussion section must be rewritten, focusing on what new this study offers compared to other studies about SRI in India.

- Please make the changes needed based on the instructions for authors and the template given by the Journal. All tables must be improved. Also, some parts of the text do not follow the rules set by the Journal. In some cases, English language must be improved. 

- Please re-write discussion based on the results/findings of your study. 

- Introduction and conclusions must also be improved.

Detailed comments:

- Line 74: In addition, direct seeded rice (DSR) systems have been considered a sustainable strategy for sustainable rice production and resilience under adverse climatic conditions apart from requiring less labor and being less costly.

- Line 81: Add reference for this: "Switching from NTP to SRI methods or to DSR would certainly change the soil food web in rice ecosystems, favoring species that prefer a more aerobic soil environment."

- Line 140: Table format is not compatible with the template set by the Journal. Also, although the abbreviations used in the table have been explained in the text, it would be useful to be added just below the table to improve the reading and comprehension of the table.

- Line 140: The abbreviation for AWD (obviously “Alternate Wetting and Drying”) is not given in Table 1.

- Line 149: Same issue here with the format of Table 2. In addition, obviously something is wrong with the data of column of temperature.

- Line 179: In the text, it is mentioned you refer to AWD applications without explaining what AWD is. In addition, in discussion there is also a reference to AWD without providing the info needed.

- Line 220: Same issue with the format of Table 3. In addition, there is only one reference for each equivalent energy category and data is taken mainly only from one reference (Devasenapathy et al., 2009). Please provide more references to support the use of these values and to reduce bias or other subjective errors.

- Line 263, Figure 1: Improve image resolution and remove frame. The image seems to be distorted horizontally.

- Line 351: “The mean yields from SRI, DSR and MSRI were significantly higher than NTP in all experiments as shown in Table 4”. The statistical term “significantly” must be followed by the corresponding p-values.

- Line 353, Table 4: Please make changes to the format of the table based on the template of the Journal. Superscripts are not described below the table. 

- Line 369, Table 5: Change format. Describe what superscripts mean below the table.

- Line 378, Table 6: Same format issues.

- Line 396: “were not significant (Table 7)”. This is not clear in Table 7. Please make the proper modifications and provide p-values and descriptions for the superscripts.

- Line 407, Table 7: Same format issues. Also, font size seems to be larger than it should be.

- Line 409: “Kg CO2-eq kg−1 grain”. Should be removed?

- Line 415: Improve English.

- Line 424, Table8: Change format. Provide info for the superscripts.

- Line 445, Figure 2: Keep the same format for all four graphs. Make corrections to image caption as well. Frames are not needed.

- Line 451: The discussion section must be rewritten. I strongly recommend you follow the same structure that you have in results. Therefore, I suggest you to provide a discussion on the following: Grain yield, Water productivity, Economic productivity, Energy use efficiency, Greenhouse gas (GHG) emissions, Microbial populations and enzyme activities, Effects on soil nematode community.

- Line 474: Remove red font color. This addition seems to be confusing. Why did you add this information here? What is the connection of the given comparison with the findings of your study? I would suggest adding your results in a column (regarding the comparison between conventional transplanting and manual/mechanized SRI Transplanting) and then discuss the information given, otherwise it is not clear how this table is relevant to the rest of the discussion.

- Line 511: Remove red font color. Change the format of the table. Several other corrections are needed here.

- Line 512: Please remove “(figures in parentheses are percentages)”. Instead, either add the % symbol or change the title to “Effects and percentage change in key variables due to Manual and Mechanized SRI Transplanting compared to Conventional Transplanting”.

- Line 513: Reference is missing.

- Line 517: “relatively lower emissions of CH4”. In which table is given this? The part of the text regarding the AWD in the discussion must be re-written.

- Line 567: This is not a good ending for the above discussion, and it must be re-written. “In this study, enhanced levels of other beneficial enzymes like alkaline phosphatase and arylsulfatase were also observed with SRI methods of cultivation (38).” This is not analyzed or explained enough, and it should be presented earlier in the text. The rest of the paragraph is about future or different research, and it is not appropriate for closing the discussion section.

- Line 575 – 583: Improve English.

- Line 580: “on par”?

- Line 591: Improve image resolution in Figure 3.

- Line 607: Add newer references or replace old with newer references.

Author Response

Reviewer 2 Report (New Reviewer)

The Authors have done a wonderful research work but there are some suggestions (copy attached) which needs to be incorporated for the manuscript  be more effective 

Author Response

This manuscript is a resubmission of an earlier submission. The following is a list of the peer review reports and author responses from that submission.

Round 1

Reviewer 1 Report

I go through the manuscript entitled “ Assessing System of Rice Intensification methods along with 2 labor-saving alternatives and climate effects in India” and found the study interesting one. The authors tried to assess SRI methods and other rice crop management systems in grain yield, soil microbial populations, soil enzyme activities, methane emissions and so on. It’s a six years study and the authors provided good data and valuable results. The introduction, methods and discussion are well described. Overall, it has the potential to be published. There are some small points which would be better to consider it before further process. 

1.     It is recommended to highlight the purpose and significance of this study in the abstract section.

2.     Change “ha-1” to “ha-1” and recheck in all documents.

3.     Please add the relative reference to EUE.

4.     Optimize Fig.2-6 quality, currently these figures lack horizontal coordinates.

5.     Please add the Conclusion section.

After these considerations, the publication of the manuscript is suggested.

Author Response

I appreciate the suggestion of the reviewer.

I had attempted all the points raised by Reviewer 1

Reviewer 2 Report

Comments on “Assessing system of rice intensification methods along with labor-saving alternatives and climate effects in India”.

It is an interesting manuscript in terms of labor-saving alternatives and climate effects related to cultural practices. However, I believe that there are few areas that require some revisions below:

Methodology

1)     Field management: land preparation and planting techniques should be described in more detail, for example, hand transplanting or machine transplanting. How many rows are there for direct seeding?  When was water applied? How many cycles for AWD practices? How deep of water as compared to a continuous flooding system? How was fertilizer applied? What was the N rate and time of applications? What was the source of nitrogen (urea, AMS, liquid fertilizer N)?

2)     Water productivity: it would be better to further explain the usage  of “digital water meters” including the table of water consumption in each cultural practice of each season (wet vs dry) and compare over years. In addition, there should be descriptions on  rainfall’s impact on water usage and efficiency each year.

3)     Energy use efficiency: need more information on energy input or requirement per unit area of each cultural practice from land preparation until harvest.

4)     Greenhouse gas emissions: how many replications for gas sampling in each cultural practice? Other than two sampling times (9:00-10:00; and 15:00-16:00) in a day, how many days in a week? When did the gas sampling start (how many days after transplanting?) and when was the last day of gas sampling (how many days before harvesting?). Since row spacings in each cultural practice were varied, how many plants were in the gas chamber of each cultural practice? What is the data on line 267-275 (average from 7 years)? It would be better to compare that by season.

Results

1)     Table of yield over 7-year trials (both seasons) should be included.

2)     Labor/energy input should be compared over the experimental period.

3)     Methane and nitrous oxide should be presented in a summary Table by seasons and years.

Author Response

I thank the suggestions of the reviewer which helped to improve the manuscript

Please see the attachment comprises  of the reply to queries

Reviewer 3 Report

The authors are attempting to broadly evaluate yield, water productivity, GHG emissions, economy, and energy efficiency for four different rice crop management systems (SRI, MSRI, DSR, NTP). However, this manuscript is very difficult to understand because of the wide range of research subjects, and lack of explanations and scientific evidence for any of the studies. For example, there are many unclear points in the methods of cultivation trials, and it is not even possible to confirm that trials had been conducted with random or replication. Furthermore, the number of samples for each crop management is limited and does not match, and in the case of DSR, there is only one sample that even a significant difference test cannot be performed. In other words, this trial design cannot evaluate the effect of different crop management. In addition, no calculation basis for energy efficiency or economic evaluation is provided. The number and timing of GHG gas sampling are also unknown. This is because GHG emissions affect rice growing stages and weather conditions. Moreover, In the soil microbial population and soil nematode communities for comparison between SRI and NTP, there is no data before and after the cultivation, so we cannot compare whether it is the effect of the crop management system. In addition, this manuscript seems to have a lack of review for previous related studies which makes it difficult to know what kind of new findings were obtained.

Other minor comments in Method are the following:

89-90 The description of crop management in the main text and the supplement do not match.

Table 1 is a summary, not a detail. There is no description of units. Is spacing 6 cm between plants correct in DSR?

102 The duration of the rain and dry seasons is unknown.

126-127 The description of mechanized transplanting in the text and Table 1 do not match.

135 Regarding water productivity, the measurement period for water volume is unknown.

137 “very accurate” should be deleted.

139 Is it possible to maintain the water regimes using fiber sheets?

146 How to calculate energy efficiency is unknown.

152 Unknown number and timing of GHG gas sampling.

176-196 No purpose is given as to why soil microbial populations and soil nematode communities are measured.

Table 3 Table needs improvement. There are notations of the B: C ratio and BCR. No explanation for other abbreviations.

Author Response

I thank the critical comments of the Reviewer. I attempted all the suggestions.

Round 2

Reviewer 2 Report

Methodology was slightly improved, however, the illogical data for greenhouse gas emissions was presented in Table 5. Author(s) described that the alternate wetting and drying (AWD) was practiced in SRI, MSRI, and DSR cultivation, meanwhile continuous flooding was implemented in the NTP cultivation. Under anaerobic conditions or low soil redox potential in NTP practice would induce microorganisms to generate significantly higher amount of methane than in the AWD water management environment. In contrast, the cycles of aerobic/anaerobic conditions in AWD would enhance significantly higher nitrous oxide emissions than the continuous flooding condition in NTP practice. Therefore, the emission levels of both methane and nitrous oxide between NTP and the other three practices would not be at the same level.

Author Response

"Please see the attachment "

Thank you for your suggestions 

Reviewer 3 Report

I can't read letters and numbers in Figs. 2-5, or it should be deleted because it overlaps with Tables 3-5. Moreover, Abbreviations such as SEm± should be explained in the annotation. 

Author Response

(The authors gave the same response as above.)
